# Touch and Go: Learning from Human-Collected Vision and Touch

**Fengyu Yang**[1*]    **Chenyang Ma**[1*]    **Jiacheng Zhang**[1]
**Jing Zhu**[1]    **Wenzhen Yuan**[2]    **Andrew Owens**[1]

[1]University of Michigan    [2]Carnegie Mellon University

https://touch-and-go.github.io

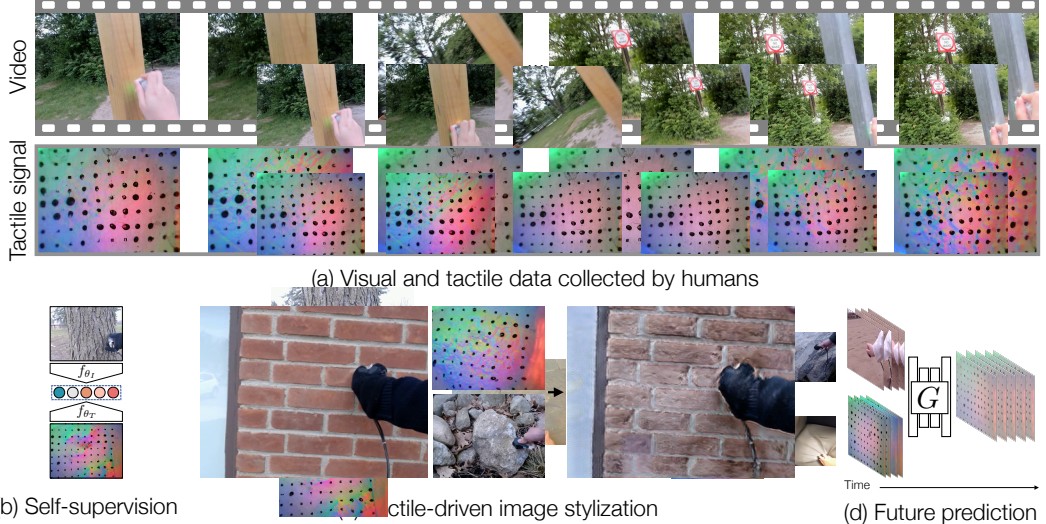

Figure 1: **The *Touch and Go* dataset.** We collect a dataset of real-world visual and touch data. (a) Humans walk through a large number of scenes, probing objects around them with a touch sensor and recording video. We apply this dataset to: (b) learning tactile features through self-supervision by associating touch with sight, (c) manipulating an image to match the tactile signal (e.g., restyling a smooth surface to match the tactile signal for a rough rock, whose photo we show for reference), (d) predicting future tactile signals from visuo-tactile inputs.

## Abstract

The ability to associate touch with sight is essential for tasks that require physically interacting with objects in the world. We propose a dataset with paired visual and tactile data called *Touch and Go*, in which human data collectors probe objects in natural environments using tactile sensors, while simultaneously recording egocentric video. In contrast to previous efforts, which have largely been confined to lab settings or simulated environments, our dataset spans a large number of "in the wild" objects and scenes. We successfully apply our dataset to a variety of multimodal learning tasks: 1) self-supervised visuo-tactile feature learning, 2) tactile-driven image stylization, i.e., making the visual appearance of an object more consistent with a given tactile signal, and 3) predicting future frames of a tactile signal from visuo-tactile inputs.

---

\* Indicates equal contribution

36th Conference on Neural Information Processing Systems (NeurIPS 2022) Track on Datasets and Benchmarks.

# 1   Introduction

As humans, our ability to correlate touch with sight is an essential component of understanding the physical properties of the objects around us. While recent advances in other areas of multimodal learning have been fueled by large datasets, the difficulty of collecting high-quality data has made it challenging for the community to develop similarly effective visuo-tactile models.

An intuitively appealing solution is to offload data collection to robots [8, 9, 42], which can acquire enormous amounts of data by repeatedly probing objects around them. However, this approach captures only a narrow, "robot-centric" slice of the visuo-tactile world. The data typically is limited to a specific environment (e.g., a robotics lab), and it fundamentally suffers from a chicken-and-egg problem, as the robot must *already* be capable of touching and manipulating the objects it acquires data from. In practice, this often amounts to recording data from tabletops, typically with small objects the robots can safely grasp. Recent work has also turned to simulation [21, 22], such as by modeling special cases where tactile interactions can be accurately simulated (e.g., rigid objects). Yet this approach, too, is highly limited. Real objects squish, deform, and bend in complex ways, and their seemingly simple surfaces can hide complicated microgeometry, such as weaves of fabric and tiny pores. Obtaining a full understanding of vision and touch, beyond simple robotic manipulation tasks, requires modeling these subtle visuo-tactile properties.

We argue that many aspects of the visuo-tactile world are currently best learned by observing physical interactions *performed by humans*. Humans can easily access a wide range of spaces and objects that would be very challenging for robots. By capturing data from objects *in situ*, the recorded sensory signals more closely match how the objects would be encountered in the wild. Inspired by this idea, we present a dataset, called *Touch and Go*, in which human data collectors walk through a variety of environments, probing objects with tactile sensors and simultaneously recording their actions on video. Our dataset spans a wide range of indoor and outdoor environments, such as classrooms, gyms, streets, and hiking trails. The objects and "stuff" [1] they contain are thus significantly more diverse than those of existing datasets, making it well-suited to self-supervised learning, and to tasks that require an understanding of material properties, such as visual synthesis tasks.

We apply our dataset to a variety of multimodal learning tasks. First, we learn tactile features through self-supervised learning, by training a model to associate images with touch. We find that the learned features significantly outperform supervised ImageNet [15] features on a robotic manipulation task, and on recognizing materials in our dataset (Fig. 1b). Second, we propose a novel task of *tactile-driven image stylization*: making an image "feel more like" a given tactile input. To solve this problem, we adapt the recent method of Li et al. [41] to generate an image whose structure matches an input image but whose style is likely to co-occur with the given tactile information. This task evaluates the ability to learn cross-modal associations – i.e., how an object feels from how it looks and vice versa. The resulting model can successfully change the texture of an input image, such as by adding bumps to a smooth surface to match the tactile information recorded from a rock (Fig. 1c). Finally, we study multimodal models for *future touch prediction*: predicting future frames of a touch sensor's recording, given both visual and tactile signals. We show that visual information improves these predictions over touch alone (Fig. 1d).

# 2   Related Work

**Simulated vision and touch.**   A variety of methods have simulated low-dimensional Biotac [59] and fabric-based tactile sensing [46]. Other work has proposed to simulate high-dimensional tactile data, based on visual tactile sensors such as GelSight [73, 32, 31]. Wang et al. [65] simulated visual and tactile data for robotic grasps of rigid objects. Gao et al. [21, 22] proposed a dataset of simulated visual, tactile, and audio data, derived from CAD models with only rigid deformation. In contrast to these works, we collect our data from *real* objects and scenes, which contain non-rigid deformation, microgeometry, and wider variations in visual appearance.

**Robotic vision and touch.**   Researchers have proposed a variety of methods that use visual and touch signals for robotic applications [9, 8, 13, 34, 37, 48, 43, 12, 70, 11, 18]. Several of these have proposed visuo-tactile datasets. Calandra et al. created a dataset for multimodal grasping [9] and regrasping [8] with a robotic arm. Li et al. [42] collected data from a robotic arm synthesis, and proposed a model based on generative adversarial networks [24] for cross-modal translation. Murali

et al. [48] proposed a dataset for tactile-only grasping. These datasets have largely been confined to specific environments (e.g., a lab space containing the robots), and only contain objects provided to them by humans that the robots are capable of interacting with. Consequently, they contain a small number of object instances (each less than 200).

**Human-collected multimodal data.** We take inspiration from work that collects data by having humans physically interact with objects *in situ*. Song et al. [60] proposed a human-collected grasping dataset. In contrast, our focus is on having humans collect rich multimodal sensory data that is well-suited to self-supervised learning. Our approach is similar to Owens et al. [51], which learns audio-visual associations by probing objects with a drumstick [51]. In contrast, we collect touch instead of sound, and record data in an approximately egocentric manner as they move from object to object. Later work predicts the trajectory of a bounced ball [55]. Sundaram et al. [61] proposed a glove that records tactile signals, and collected a dataset of human grasps for 26 objects in a lab setting. Other work predicts hand pose from touch [75]. Burka et al. combined several haptic sensors [5, 6]. They then demonstrated the resulting sensor by collecting a preliminary (currently unreleased) dataset of 357 real-world surfaces, and training a model to predict human ratings of 4 surface properties from touch. By contrast, we have significantly larger and more diverse data from indoor and outdoor scenes (rather than flat, largely indoor surfaces), use a rich vision-based tactile sensor (GelSight), and demonstrate our dataset on cross-modal prediction tasks.

**Multimodal feature learning.** In seminal work, de Sa [14] proposed to learn from correlating sight from sound. A variety of methods of been proposed for training deep networks to learn features from audio-visual correlations [49, 51, 52, 2, 35, 50, 47], from images and depth [63], and from vision and language [33, 45, 17, 56], and matching images and touch [38]. We adapt the contrastive model of Tian et al. [63] to visuo-tactile learning.

**Multimodal image prediction.** A variety of methods have been proposed for predicting images from another modality, such as by using text or labels [58, 30, 4, 57] or sound [39, 10, 41]. Li et al. [41] proposed a model for audio-driven stylization, i.e. learning to restyle images to better match an audio signal. We adapt this model to tactile-driven stylization, creating a model that is conditioned on tactile inputs instead of sound. We also take inspirations from work on future video prediction [16, 23, 54, 66, 20, 71, 67, 3, 64]. In particular, Tian et al. [62] trains an action-conditioned video prediction method to estimate future tactile signals, using a GelSight sensor controlled by a CNC machine. In contrast, we predict future tactile signals from natural objects, and show that visual information can improve the prediction quality.

**Cross-modal image stylization.** Many areas of multimodal perception have used cross-modal image stylization to evaluate whether models can capture associations between modalities (e.g. Text-to-image stylization, Audio-visual stylization). We adapt the cross-modal stylization method of Li et al. [41] to tactile-driven stylization, a task that requires learning visual-tactile associations – i.e., how an object feels from how it looks and vice versa. This direction is also related to work in computer graphics that synthesizes images that have specific material properties [40, 25, 26, 77]. In contrast to these works, we synthesize images with material properties that are captured implicitly from a touch signal.

## 3  The *Touch and Go* Dataset

We collect a dataset of natural vision-and-touch signals. Our dataset contains multimodal data recorded by humans, who probe objects in their natural locations with a tactile sensor. To more easily train and analyze models on this dataset, we also collect material labels and identify the frames within the press.

### 3.1  Collecting a natural visuo-tactile dataset

To acquire our dataset, human data collectors (the authors) walked through a variety of environments, probing the objects with a tactile sensor. To obtain images that show clear, zoomed-in images of the objects being touched, two people collected data at once: one who presses the tactile sensor onto an object, and another who records an "approximately egocentric" video (see supplement for a visualization) of their hand. The two data collectors moved from object to object in the space as part of a single, continuously recording video, touching the objects around them. We show examples from our dataset in Fig. 2. The captured data varies heavily in material properties (e.g., soft/hard, smooth/rough), geometries, and semantics.

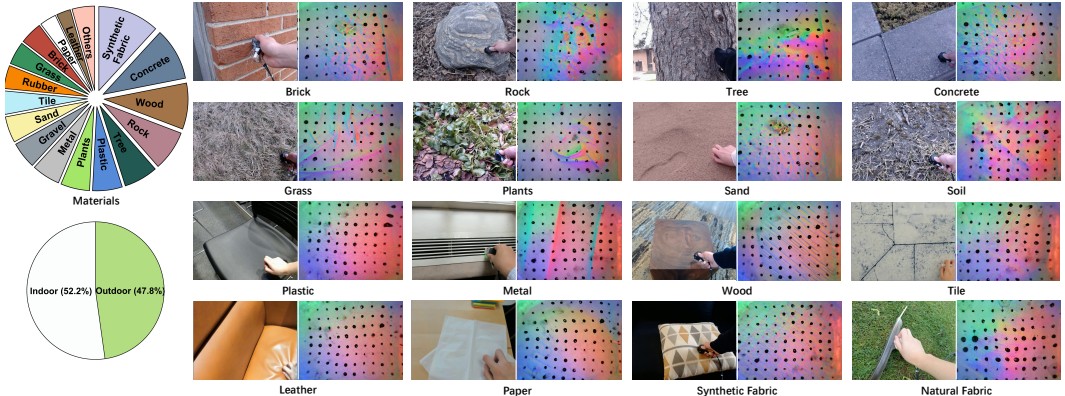

Figure 2: **The *Touch and Go* Dataset**. Human data collectors record paired visual and tactile information by probing objects in a variety of indoor and outdoor spaces. We show a selection of images, paired with the corresponding frame recorded by the GelSight tactile sensor. We show 16 representative categories (out of 20), and provide the distribution of material and scene types.[2]

**Capturing procedure.** To ensure that our dataset captures the natural variation of real-world vision and touch, we collect both rigid and deformable objects in indoor and outdoor scenes. These scenes include rooms in university buildings, such as classrooms and hallways, apartments, hiking trails, playgrounds, and streets. We show example footage from our model in Fig. 1, and in the supplement. The data collectors selected a variety of objects in each scene to press including chairs, walls, ground, sofa, table, etc. in indoor scenes, and grass, rock, tree, sand etc. in outdoor scenes, pressing each one approximately 3 times. Each press lasted for 0.7 sec on average. If an object contains multiple materials (e.g., a chair with a cushion and a plastic arm), collectors generally directed their presses to each one. The data collectors also aimed to touch object parts with complex geometry, rather than flat surfaces. To avoid capturing human faces, in public spaces the captures point the camera toward the ground when moving between objects. Since the GelSight may provide information about force implicitly [73], and explicit force readings are not required for many visuo-tactile tasks, we do not use a separate force sensor.

**Hardware.** For the tactile sensor, we use GelSight [32], the variant designed for robotic manipulation [73]. This is a vision-based tactile sensor, approximately 1.5cm in diameter, in which a camera observes the deformation of a curved elastomer gel illuminated by multiple colored light sources, with markers embedded inside it. When the sensor is pressed against an object, the gel deforms, which results in changes to the reflected illumination. The color conveys the surface normal of the object being touched, similar to photometric stereo. These black dots are "markers" that are physically embedded within the GelSight's elastomer gel. Thus, GelSight records a video in which surface orientation, depth, and shear can be estimated by analyzing the appearance of each video frame. The tactile sensor's recordings are recorded concurrently with visual images from an ordinary webcam (both at approximately 27 Hz). Both videos (tactile and visual) are recorded on a single computer.

### 3.2 Annotating the dataset

To make it easier to analyze results and train models, we provide annotations for material categories and frames within the press.

**Detecting the press.** As the data collectors move between objects, the tactile sensor does not make contact with anything. Thus, for convenience, we provide the subset of frames within the touch, so that applications can use trimmed videos (Sec. 4). To obtain these timings, we train a detector for press detection. We (the authors) hand-label 10k frames and train a binary ResNet-18 classifier [27] that operates on GelSight images. On our test set of 2k hand-labeled frames, our classifier obtains 97% accuracy (chance is around 66%). To further ensure high accuracy, we hire workers to review the frames within the touch and correct errors.

---

[2]Following common practice, the tactile images are enhanced for visualization purpose. Contrast and sharpness are increased by 30% and saturation is increased by 20%.

Table 1: **Tactile datasets.** We compare attributes of our dataset with several previously proposed datasets.

| | Object inst. | Touches | Source | Real-world | Environment | Sensor |
|---|---|---|---|---|---|---|
| More Than a Feeling [8] | 65 | 6.5k | Robot | ✓ | Tabletop | GelSight [72] |
| The Feeling of Success [9] | 106 | 9.3k | Robot | ✓ | Tabletop | GelSight [72] |
| VisGel [42] | 195 | 12k | Robot | ✓ | Tabletop | GelSight [72] |
| ObjectFolder 1.0 [21] | 100 | - | Synthetic | ✗ | - | DIGIT [36] |
| ObjectFolder 2.0 [22] | 1000 | - | Synthetic | ✗ | - | GelSight [72] |
| Burka et al. [7] | 357 | 1.1k | Human | ✓ | Mostly Indoor | Multiple Sensors |
| **Touch and Go (Ours)** | 3971 | 13.9k | Human | ✓ | Indoor/Outdoor | GelSight [72] |

**Labeling materials.** We label the material category for all (visual) video frames that our detector predicts within the press, using a labeling scheme similar to [51]. Online workers assign a label from a list of categories. If an object is not in this category list, the workers will label it *other material*. To ensure accuracy, we have 5 workers label each image. We show the distribution of labels in Fig. 2.

## 3.3 Dataset analysis

We analyze the contents of our dataset and compare it to other works.

**Data distribution.** In Fig. 2, we show statistics of labeled materials and scene types, and provide qualitative results from the dataset. It contains approximately 13.9k detected touches and approximately 3971 individual object instances. Since we do not explicitly label instances, we obtain the latter number by exhaustively counting the objects in 10% of the videos and extrapolating. Our dataset is relatively balanced between indoor (52.2%) and outdoor (47.8%) scenes. We found that several categories, namely *synthetic fabric*, *tile*, *paper*, and *leather*, are only present in our indoor scenes, while *tree*, *grass*, *plant*, and *sand* are only present in outdoor scenes. The remaining materials exist in both scenes. We provide more details in the supplement.

**Comparison to other datasets.** In Table 1, we compare our dataset to several previously proposed visuo-tactile datasets collected by robots, by humans, or through simulation. Our dataset contains approximately $4\times$ as many object instances as the second-largest dataset, the simulation-based ObjectFolder 2.0 [22], and $11\times$ larger compared with the human-collected dataset by Burka et al. [7]. Compared to the existing robot-collected datasets, ours contains more touches (e.g., $1.15\times$ more than VisGel [42] and $1.5\times$ more than The Feeling of Success [9]). Our dataset also contains data from more diverse scenes than prior work, with a mixture of natural indoor and outdoor scenes. In contrast, robot-collected datasets [8, 9, 42] are confined to a single lab space containing the robot.

**Qualitative comparison to other datasets** We show qualitative examples of data from other datasets in Fig. 3 to help understand the differences between our dataset and those of previous work: Object Folder 2.0 [22], which contains virtual objects, and two robotic datasets: Feeling of Success [9], and VisGel [42]. We show examples from indoor scenes, since the other datasets do not contain outdoor scenes, and with rigid materials (since the virtual scenes do not contain deformable materials). Each row illustrates objects which are composed of similar materials, along with their corresponding GelSight images. As can be seen, the robot-centric datasets [42, 9] are confined to a fixed space. Their objects are also smaller than those in our dataset, since they must be capable of being grasped by the robot's gripper. Synthetic datasets [21, 22] do not contain complex microgeometry, and their rigid objects do not deform when pressed.

## 4 Applications

To evaluate the effectiveness of our dataset, we perform tasks that are designed to span a variety of application domains, including representation learning, image synthesis, and future prediction.

### 4.1 Multimodal self-supervised representation learning

We ask, first, whether we can use the multimodal data to learn representations for the tactile modality by associating touch with sight. We then ask how well the learned representations convey material properties from our dataset, and whether they are useful for robotic learning tasks whose data has been collected in other works [8, 9]. The latter task requires a significant amount of generalization, since the objects manipulated by robots while our dataset is collected by humans.

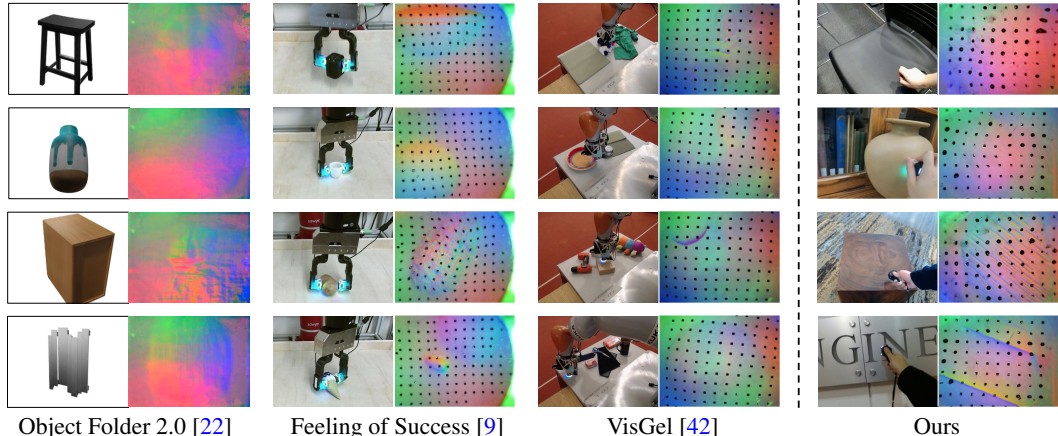

| Object Folder 2.0 [22] | Feeling of Success [9] | VisGel [42] | Ours |

Figure 3: **Visuo-tactile data from other datasets**. We provide qualitative examples of visual and tactile data from other datasets (left), along with examples from similar material taken from our dataset (right).

Our goal is to learn a representation that captures the information and correspondences between visual and tactile images, which can be useful for downstream tasks. Given the visual and tactile datasets, $X_I$ and $X_T$, we aim to extract the corresponding visual-tactile pairs, $\{\mathbf{x}_I^i, \mathbf{x}_T^i\}$ and mismatched pairs $\{\mathbf{x}_I^i, \mathbf{x}_T^j\}$ using the Contrastive Multiview Coding (CMC) model proposed by Tian et al. [63]. The detailed procedure is shown below.

For each visual-tactile image pair, we first encode visual images and tactile inputs as L2-normalized embeddings using two networks, where $z_I = f_{\theta_I}(\mathbf{x}_I)$ for visual images and $z_T = f_{\theta_T}(\mathbf{x}_T)$ for tactile images. Recall that our goal is to find the corresponding sample of the other modality, given a set $S$ that contains both the corresponding example and $K-1$ random examples. When matching a visual example $\mathbf{x}_I^i$ to tactile, the loss for matching visual images to tactile images is:

$$\mathcal{L}_{\text{contrast}}^{X_I,X_T} = -\log \frac{\exp(f_{\theta_I}(\mathbf{x}_I^i) \cdot f_{\theta_T}(\mathbf{x}_T^i)/\tau)}{\sum_{j=1}^{K} \exp(f_{\theta_I}(\mathbf{x}_I^i) \cdot f_{\theta_T}(\mathbf{x}_T^j)/\tau)} \tag{1}$$

where $\tau = 0.07$ is a constant and $j$ indexes the tactile examples in $S$. Analogously, we can obtain a loss in which tactile examples are matched to images, $\mathcal{L}_{\text{contrast}}^{X_T,X_I}$. We minimize both losses:

$$\mathcal{L}(X_I, X_T) = \mathcal{L}_{\text{contrast}}^{X_I,X_T} + \mathcal{L}_{\text{contrast}}^{X_T,X_I} \tag{2}$$

The training details and hyperparameters are provided in the supplementary material.

### 4.2 Tactile-driven image stylization

Touch provides complementary information that may not be easily conveyed through other modalities that are commonly used to drive image stylization, such as language and sound. For example, touch can precisely define how smooth/rough a surface ought to be, and express the subtle shape of its microgeometry. A model that can successfully predict these properties from visuo-tactile data therefore ought to be able to translate between modalities. Inspired by the audio-driven image stylization of Li et al. [41], we propose the task of *tactile-driven image stylization*: making an image look as though it "feels like" a given touch signal.

Following Li et al. [41], we base our approach on contrastive unpaired translation (CUT) [53]. Given an input image $\mathbf{x}_I$ and a tactile example $\mathbf{x}_T$, our goal is to manipulate the image via a generator such that the manipulated pair $\hat{\mathbf{x}}_I = G(\mathbf{x}_I, \mathbf{x}_T)$ is more likely to co-occur in the dataset. Our model consists of an image translation network that is conditioned on a tactile example (a GelSight image). The loss function encourages the model to preserve the image structure, which is enforced using an image-based contrastive loss [53], while adjusting the image style such that the resulting textures are more likely to co-occur with the tactile signal, which is measured using a visuo-tactile discriminator.

**Making sight consistent with touch.** We train the model with a discriminator $D$ that distinguishes between real (and fake) visuo-tactile pairs. During training time, we shuffle the dataset to generate the

set $S_n$ containing mismatched image-tactile pairs $\{\mathbf{x}_I, \mathbf{x}'_T\} \in S_n$. Likewise, we define the original dataset as $S_m$ which contains matched image-tactile pairs $\{\mathbf{x}_I, \mathbf{x}_T\} \in S_m$. In formal terms, the visual-tactile adversarial loss can be written as:

$$\mathcal{L}_{\mathrm{VT}} = \mathbb{E}_{\{\mathbf{x}_I, \mathbf{x}_T\} \sim S_m} \log D(\mathbf{x}_I, \mathbf{x}_T) + \mathbb{E}_{\{\mathbf{x}_I, \mathbf{x}'_T\} \sim S_n} \log(1 - D(G(\mathbf{x}_I, \mathbf{x}'_T), \mathbf{x}'_T)) \qquad (3)$$

**Loss.** We combine our visuo-tactile discriminator with the structure-preserving loss used in Li et al. [42], which was originally proposed by CUT [53]. This loss, which we call $\mathcal{L}_{\mathrm{CUT}}$, works by training a contrastive learning model that puts patches in the input and predicted images into correspondence, such that patches at the same positions are close in an embedding space. Please see the supplement for more details. The overall loss is:

$$\mathcal{L}_{\mathrm{TDIS}} = \mathcal{L}_{\mathrm{VT}} + \mathcal{L}_{\mathrm{CUT}}. \qquad (4)$$

### 4.3 Multimodal future touch prediction

Inspired by the challenges of action-conditional future touch prediction [62], we use our dataset to ask whether *visual* data can improve our estimates of future tactile signals: i.e., what will this object feel like in a moment? Visual information can help touch prediction in a number of ways, such as by conveying material properties (e.g., deformation) and and geometry. It can also provide information about which action is being performed. One may thus also consider it as an *implicit* form of action conditioning [19], since the visual information provides information analogous to actions.

We adapt the video prediction architecture of Geng et al. [23] to this task. This model is based on residual networks [27, 29] with 3D space-time convolutions (please see the supplement for architecture details). We predict multiple frames by autoregressively feeding our output images back to the original model. Given a series of paired visual and tactile images from times 1 to $t$, $\{(\mathbf{x}_I^1, \mathbf{x}_T^1), ..., (\mathbf{x}_I^t, \mathbf{x}_T^t)\}$, our goal is to predict the subsequent tactile image, $\hat{\mathbf{x}}_T^{(t+1)}$. We train the model with L1 and perceptual loss, following [23].

## 5 Experiments

### 5.1 Self-supervised feature learning

We train a self-supervised model that learns to associate images with touch. We evaluate this learned representation on two downstream tasks: robot grasping and material understanding tasks. .

**Robotic grasping task.** For the robot grasping task, we use the experimental setup and dataset of Calandra et al. [9]. Thus, the task requires generalizing to data recorded in a very different environment, and with different GelSight sensors. The goal of this task is to predict whether a robotic arm will successfully grasp an object, based on inputs from two GelSight images recorded before and after grasping. Since there is no standard training/test split from [9], we split their objects randomly into training/test. The resulting dataset contains 68 objects and 5921 touches for training, 16 objects and 1204 touches for validation, and 21 objects and 1204 touches for testing. Similar to the tactile-only model from Calandra et al. [9], we compute features for each of the 4 tactile images (before/after images for 2 tactile sensors) using our self-supervised model. We concatenate these features together and train a linear classifier to solve this binary classification task.

**Material understanding tasks.** We evaluate whether the learned features convey material properties. Given tactile features, we recognize: 1) material categories, 2) hard vs. soft surfaces, and 3) smooth vs. rough surfaces. Following [51], we re-categorize material categories to generate soft/hard labels and hire online workers to label the smooth/rough according to the visual image. Since the smoothness and roughness may vary within a material category, we hire online workers to label the smooth vs. rough according to the visual image. To avoid providing our self-supervised learning model with object instances that appear in the linear probing experiment, we split the dataset into an unlabeled set containing 5172 touches (51.7%), a labeled training set of 3921 touches (39.2%), and a labeled test set of 923 touches (9.1%). We split the dataset by video (rather than by frame) to avoid having the same (or nearly same) object appear in both training and test.

Table 2: Comparison of pretrained models on ImageNet and other tactile datasets on different downstream tasks. We also evaluate variations of our model trained only on subsets of the material classes.

| Dataset | Method | Grasping Acc(%) | Material Acc(%) | Hard/Soft Acc (%) | Rough/Smooth Acc (%) |
|---------|--------|-----------------|-----------------|-------------------|----------------------|
| Chance | - | 56.1 | 18.6 | 66.1 | 56.3 |
| ImageNet [15] | Supervised | 73.0 | 46.9 | 72.3 | 76.3 |
| Object Folder 2.0 [22] | Visuo-tactile CMC | 69.4 | 36.2 | 72.0 | 69.0 |
| VisGel [42] | Visuo-tactile CMC | 75.6 | 39.1 | 69.4 | 70.4 |
| Ours - 25% Classes | Visuo-tactile CMC | 62.3 | 25.7 | 67.3 | 65.3 |
| Ours - 50% Classes | Visuo-tactile CMC | 66.5 | 35.9 | 71.2 | 66.8 |
| Ours - 75% Classes | Visuo-tactile CMC | 70.8 | 48.4 | 73.3 | 74.7 |
| Ours - 100% Classes | Visuo-tactile CMC | **78.1** | **54.7** | **77.3** | **79.4** |

**Implementation details.** We train our model for 240 epochs, using the optimization parameters from CMC [63], after adjusting the learning rate schedule to compensate for longer training. We set the weight decay to be $10^{-4}$. We train our model with the batch size of 128 on 4 Nvidia 2080-Ti GPUs. For the downstream classification tasks, we froze our network weights and obtained visual features by performing global average pooling on the final convolutional layer. We follow the approach of [63] for learning the linear classifier.

**Comparison to other feature sets.** We show downstream classification results in Table 2. To evaluate the effectiveness of our dataset, we compare our learned features to those of several other approaches. These include using supervised ImageNet [15] features, which are commonly used to represent GelSight images [74, 9, 8], and visual CMC [63] features trained on ImageNet, which treats the L and ab color channels of an image as different modalities. We see that our model obtains significantly better performance than these models on both tasks, due to its ability to learn from real tactile data. These results suggest that our dataset provides a useful signal for training self-supervised tactile representations. We also show that increasing the material categories leading to much better downstream performance.

## 5.2 Tactile-driven image stylization

We use our model to modify the style of an image to match a touch signal.

**Implementation details.** Following Li et al. [41], during training we sample a random image from the dataset, along with a second visuo-tactile pair, and use the pair to restyle the image, using the loss in Eq. 4. We provide architectural details in the supplement. For the discriminator we adopt the PatchGAN architecture [28]. The architecture of discriminator follows [41], which concatenates the two input images channel-wise, and passes the combined images to the discriminator. We train our model on 4 Nvidia 2080-Ti GPUs for 100 epochs with a batch size of 8 and the learning rate of 0.0002. We augment the vision images with random cropping and horizontal flipping.

**Experimental setup.** Following [41], we evaluate our model by restyling images in our dataset with random tactile inputs, both of which are taken from the test set. We use evaluation metrics that measure the consistency of the manipulated image with the example used for conditioning. First, we measure the similarity of the manipulated image and the tactile example used for conditioning. Similar to [41], we use our trained CMC model, by taking the dot product between visual and tactile

Table 3: Quantitative results for tactile-driven image stylization.

| Method | CMC | Material |
|--------|-----|----------|
| Baseline | 0.165 | 0.107 |
| CycleGAN [78] | 0.178 | 0.129 |
| Ours | **0.197** | **0.142** |

embeddings. Second, we compare material prediction consistency between the manipulated image with the (held out) conditional image. We use our material classifier to categorize the predicted and conditioning images, and measure the rate at which they agree. Since this is a novel task, we create a second variation of our model for comparison, following [78]. This model performs the stylization using CycleGAN, rather than CUT (see supplement for details).

**Quantitative results.** Quantitative results are shown in Table 3. Here, the "baseline" indicates results from original image before stylization. We can see that the CUT-based method obtains higher

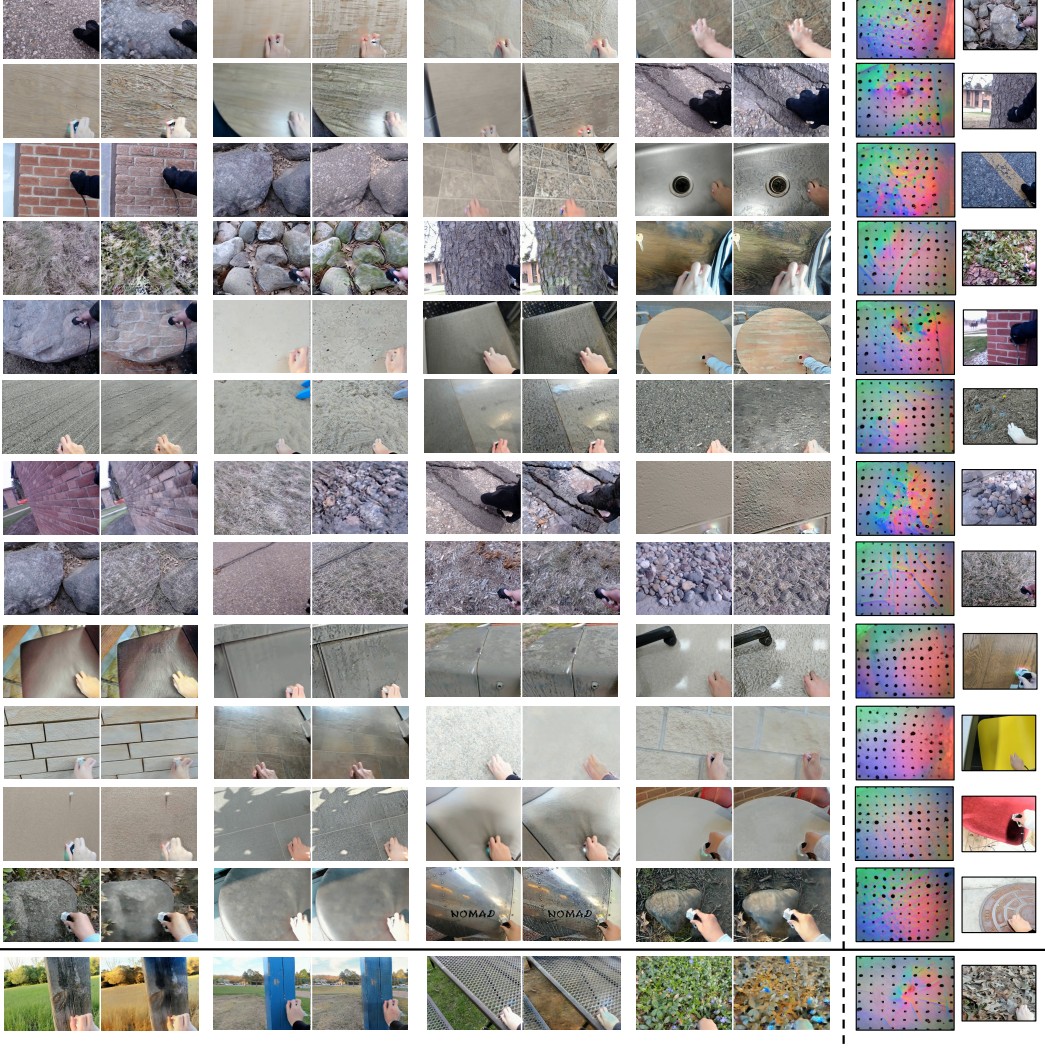

Figure 4: Qualitative results of our model on tactile-driven image stylization. For each row, we show an input image (left) and the manipulated image (to its right) obtained by stylizing with a given tactile input (right side). For reference, we also show the image that corresponds to the tactile example at rightmost (not used by the model). The manipulated images convey physical properties of the tactile signal, such as its roughness (e.g., first three rows) or smoothness (e.g., row 10). Other inputs result in images that combine the properties of two inputs (e.g., by adding grass, as in row 8). We also show failure cases in the last row. *Zoom in for better view.*

CMC similarity than a CycleGAN-based method [78]. In terms of material classification consistency, our model consistently outperforms CycleGAN-based method [78].

**Qualitative results.**    In Fig. 7, we show results from our model. Our model successfully manipulates images to match tactile inputs, such as by making surfaces rougher or smoother, or by creating "hybrid" materials (e.g., adding grass to a surface). These results are obtained *without* having access to the tactile example's corresponding image, suggesting that the model has learned which physical properties are shared between sight and touch.

### 5.3   Multimodal video prediction

We evaluate our model for predicting future tactile signals. We compare a tactile-only model to a multimodal visuo-tactile model, and show that the latter obtains better performance.

**Experimental setting.**    We evaluate the effectiveness of multimodal inputs using three context frames to predict the next frame under two different time horizons: skipping 3 and 5 frames between

Table 4: Quantitative results for video prediction.

| Time horizon | Method | Modality | L1 ↓ | SSIM ↑ | LPIPS ↓ |
|---|---|---|---|---|---|
| Skip 3 frames | SVG [16] | Touch only | 0.782 | 0.572 | 0.391 |
| | Geng et al. [23] | Touch only | 0.628 | 0.708 | 0.103 |
| | SVG [16] | Touch + vision | 0.757 | 0.602 | 0.368 |
| | Geng et al. [23] | Touch + vision | **0.617** | **0.719** | **0.091** |
| Skip 5 frames | SVG [16] | Touch only | 0.807 | 0.513 | 0.412 |
| | Geng et al. [23] | Touch only | 0.691 | 0.698 | 0.279 |
| | SVG [16] | Touch + vision | 0.762 | 0.546 | 0.397 |
| | Geng et al. [23] | Touch + vision | **0.663** | **0.713** | **0.265** |

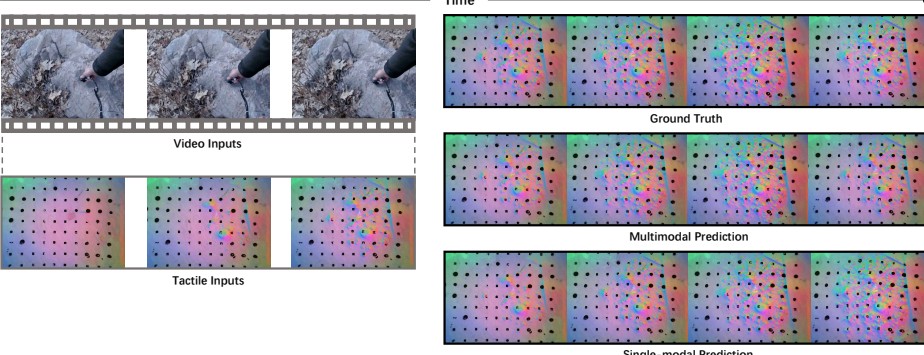

Figure 5: **Future touch prediction.** We show the results of tactile-only and visuo-tactile models.

contexts. Following [23], we adopt three evaluation metrics: MAE, SSIM [69] and LPIPS [76]. We provide training hyperparameters in the supplement.

**Multimodal vs. single-modal prediction.** We show the quantitative results in Table 4. Here, we adopt two video prediction baselines [16] and [23]. We can see that, by incorporating our dataset's visual signal, the models gain a constant performance increase under different evaluation metrics for both model, under both experimental settings. The gap becomes larger for longer time horizon, suggesting that visual information may be more helpful in this case.

## 6 Discussion

We proposed *Touch and Go*, a human-collected visuo-tactile dataset. Our dataset comes from real-world objects, and is significantly more diverse than prior datasets. We demonstrate its effectiveness on a variety of applications that involve robotic manipulation, material understanding, and image synthesis. In the tradition of previous work [51], we see our work as a step toward *human-collected* multimodal data collection, in which humans equipped with multiple sensors collect diverse dataset by recording themselves physically interacting with the world. We hope this data will enable researchers to study diverse visuo-tactile learning applications, beyond the "robotics-centric" domains that are often the focus of previous efforts.

**Limitations.** Collecting diverse tactile data is an ongoing challenge, since it requires physically being present in the locations where data is collected. While adding human collectors improves diversity in many ways, our dataset was mainly collected in one geographic location (near University of Michigan's campus). Consequently, the data we recorded may not generalize to all spaces. The use of humans in the data collection process also potentially introduces bias, which differs from "robotic" or "virtual data" bias. For example, humans may choose unrepresentative parts of the objects to probe, and do not perform actions with consistent force. The humans who recorded the dataset may also not be representative of the general population, which may introduce bias (e.g., in skin tone).

**Acknowledgements.** We thank Xiaofeng Guo and Yufan Zhang for the extensive help with the GelSight sensor, and thank Daniel Geng, Yuexi Du and Zhaoying Pan for the helpful discussions. This work was supported in part by Cisco Systems and Wang Chu Chien-Wen Research Scholarship.

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

## A Project webpage

We've provided a webpage for our dataset, which contains a link to the dataset. We also provide additional examples from our dataset (c.f., Fig 2 of the main paper).

## B Dataset file structure

Our dataset is currently available through our webpage (and directly via this link). For long-term maintenance, we will upload our dataset to University of Michigan's EECS web servers after acceptance.

The `touch_and_go` directory contains a `dataset` directory of raw videos, `extract_frame.py` that convert raw videos to frames, `label.txt` of material labels for frames within the press, and `category_reference.txt` of the name for each category in `label.txt`.

Each raw video folder in the *Dataset* folder consists of six items:

- `video.mp4`: Raw RGB video recording the interaction of human probing objects.
- `gelsight.mp4`: Raw GelSight (tactile) video for objects.
- `time1.npy`: The recording time for each frame in "`video.mp4`".
- `time2.npy`: The recording time for each frame in "`gelsight.mp4`".
- `video_frame`: The folder containing all the frames in "`video.mp4`". (Generated after running `extract_frame.py`)
- `gelsight_frame`: The folder containing all the frames in "`gelsight.mp4`". (Generated after running `extract_frame.py`)

We have provided qualitative examples of the videos on our project page. To view the videos at full resolution, please download them.

## C Egocentric recording setup

As shown in Fig. 6, we use a webcam to record the RGB video and a GelSight sensor to capture the tactile signals, which are both connected to one laptop computer. To obtain images that show clear, zoomed-in images of the objects being touched, two people collected data at once: one who presses the tactile sensor onto an object, and another who records an "approximately egocentric" video. Alternatively, one person may record both signals, while another holds the computer, providing them with a view of what they are pressing via the screen. In this way, they can ensure that the objects they are probing appear approximately in the center of the recorded images and increase the stability of the recording.

## D Category list

We conclude the objects appeared in our dataset into 20 categories according to their material property. All these categories are listed with decreasing number of quantities in terms of the number of touches. Label Num. denotes the number in the `label.txt` representing each category.

## E Implementation details for self-supervised learning

When training the contrastive multiview coding (CMC) model, we use a learning rate of 0.03 and train for 240 epochs. We use SGD as our optimizer and set the weight decay to be $10 \times 10^{-4}$ and the momentum to be 0.9. We use a batch size of 128 on 4 Nvidia 2080-Ti GPUs. For the linear probing stage in both downstream tasks, we fixed the weight of our pretrained backbone and adopt the global average pooling at the last layer followed by a linear classifier. We use a learning rate of 0.01 for ResNet-18 and 0.1 for ResNet-50. For both material classification and robot grasping, we train the linear classifiers with 60 epochs and a batch size of 256.

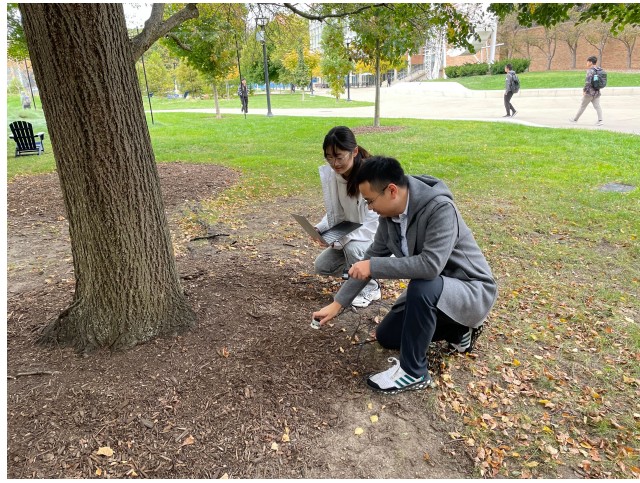

Figure 6: A photo of two humans collecting data in the wild.

| Material | Scene | Quantity | Label Num. |
|---|---|---|---|
| Synthetic Fabric | Indoor | 1.65K | 8 |
| Concrete | Indoor/Outdoor | 1.40K | 0 |
| Wood | Indoor/Outdoor | 1.24K | 3 |
| Rock | Indoor/Outdoor | 1.08K | 15 |
| Tree | Outdoor | 0.91K | 12 |
| Plastic | Indoor/Outdoor | 0.80K | 1 |
| Plants | Outdoor | 0.78K | 18 |
| Metal | Indoor/Outdoor | 0.76K | 4 |
| Gravel | Indoor/Outdoor | 0.71K | 16 |
| Sand | Outdoor | 0.70K | 17 |
| Tile | Indoor | 0.63K | 6 |
| Rubber | Indoor/Outdoor | 0.62K | 10 |
| Grass | Outdoor | 0.61K | 13 |
| Brick | Indoor/Outdoor | 0.60K | 5 |
| Paper | Indoor | 0.45K | 11 |
| Leather | Indoor | 0.38K | 7 |
| Glass | Indoor/Outdoor | 0.23K | 2 |
| Natural Fabric | Indoor/Outdoor | 0.22K | 9 |
| Soil | Indoor/Outdoor | 0.16K | 14 |
| Others | Indoor/Outdoor | 0.09K | 19 |

Table 5: We provide statistics for different material categories.

## F   Details for Tactile-driven image stylization

**Architecture.**   Our model consists of a multi-modal generator, a tactile-visual texture discriminator and a patch-wise structure discriminator. We can further break up our multi-modal generator into three components, an image encoder $G_{\text{enc\_I}}$, a tactile encoder $G_{\text{enc\_T}}$ and a decoder $G_{\text{dec}}$. Given our dataset that contains unpaired instances $S_n = \{\mathbf{x}_I, \mathbf{x}'_T\}$, the output image $\hat{\mathbf{x}}_I$ can be expressed as $\hat{\mathbf{x}}_I = G(\mathbf{x}_I, \mathbf{x}'_T) = G_{\text{dec}}(\text{concat}(G_{\text{enc\_I}}(\mathbf{x}_I), G_{\text{enc\_T}}(\mathbf{x}'_T)))$.

**Structure preserving loss ($\mathcal{L}_{\text{CUT}}$).**   Our goal in this tactile-guided image stylization is to restyle the source image with the textures that are associated with the target tactile input while preserving the source structure. Following previous approaches [53, 41], we introduce an a noise contrastive estimation (NCE) loss [53] on the image encoder $G_{\text{enc\_I}}$ that helps preserve the structural information between the visual input $\mathbf{x}_I$ and the generated image $\hat{\mathbf{x}}_I$.

This loss is motivated by recent contrastive learning to maximize the probability for the neural network to select the corresponding patch in both the original image $\mathbf{x}_I$ and the generated image $\hat{\mathbf{x}}_I$. Specifically, we select a query patch from the generated $\hat{\mathbf{x}}_I$, one positive patch and $N$ negative patches from the original image $\mathbf{x}_I$. Then we encode these patches into a $K$ dimensional vectors by a

MLP so that query vector $\boldsymbol{q}$, positive vector $\boldsymbol{v}^+$ belong to $\mathbb{R}^K$ and negative vectors $\boldsymbol{v}^- \in \mathbb{R}^{N \times K}$:

$$l(\boldsymbol{v}, \boldsymbol{v}^+, \boldsymbol{v}^-) = -\log \frac{\exp(\frac{\boldsymbol{q} \cdot \boldsymbol{v}^+}{\tau})}{\exp(\frac{\boldsymbol{q} \cdot \boldsymbol{v}^+}{\tau}) + \sum_{n=1}^{N} \exp(\frac{\boldsymbol{q} \cdot \boldsymbol{v}^-}{\tau})} \tag{5}$$

where $\tau$ is the temperature parameter.

Since our image encoder is a multi-layer convolutional network, we take advantage of multiple feature stacks generated from different layers. Specifically, we select $L$ layers of feature stacks and pass them into a MLP $M$ and the output is $M(G^l_{\text{enc\_I}}(\mathbf{x}_I)) = \{\boldsymbol{v}_l^1, \boldsymbol{v}_l^2, ..., \boldsymbol{v}_l^N, \boldsymbol{v}_l^{N+1}\}$, where $l \in \{1, ..., L-1, L\}$. Here, we denotes $G^l_{\text{enc\_I}}(\mathbf{x}_I)$ as the feature stacks at layer $l$. Similarity, we apply this to the generated image $\hat{\mathbf{x}}_I$ so that we get our query vector for each layer, which can be represented as $\{\boldsymbol{q}_l^1, \boldsymbol{q}_l^2, ..., \boldsymbol{q}_l^N, \boldsymbol{q}_l^{N+1}\}$. Thus, for each sample index $n$ at layer $l$, we let $\boldsymbol{v}_l^n \in \mathbb{R}^{N \times C_l}$ as the positive samples and other features $\boldsymbol{v}_l^{(N+1) \backslash n} \in \mathbb{R}^{N \times C_l}$ as negative samples, where $C_l$ indicates the channel of the layer $l$. Thus our multi-layer NCE loss can be represented as the following:

$$\mathcal{L}_{\text{CUT}} = \mathbb{E}_{\mathbf{x}_I \sim S_n} \sum_{l=1}^{L} \sum_{n=1}^{N+1} l(\boldsymbol{q}_l^n, \boldsymbol{v}_l^n, \boldsymbol{v}_l^{(N+1) \backslash n}) \tag{6}$$

where $S_n$ contains mismatched image-tactile pairs $\{\mathbf{x}_I, \mathbf{x}'_T\}$, as defined in our main text.

**Implementation details.** Our image encoder and decoder of the generator are fully convolutional neural networks consisting of 9 blocks of ResNet-based CNN bottlenecks. The first convolution layer is set to $7 \times 7$ and the rest are set to $3 \times 3$. For the tactile encoder, we adopt a ResNet-18 [27] backbone pretrained on the ImageNet [15]. For the discriminator we adopt the PatchGAN architecture [28]. To compute the NCE loss, we extract features from five different layers: the input image layer, the first and second downsampling convolution layer and the first and fifth residual blocks. We train our model on 4 Nvidia 2080-Ti GPUs for 100 epochs with the batch size of 8 and learning rate of 0.0002. For input visual images, we use a random crop and horizontal flip.

## G More results for tactile-drive image stylization

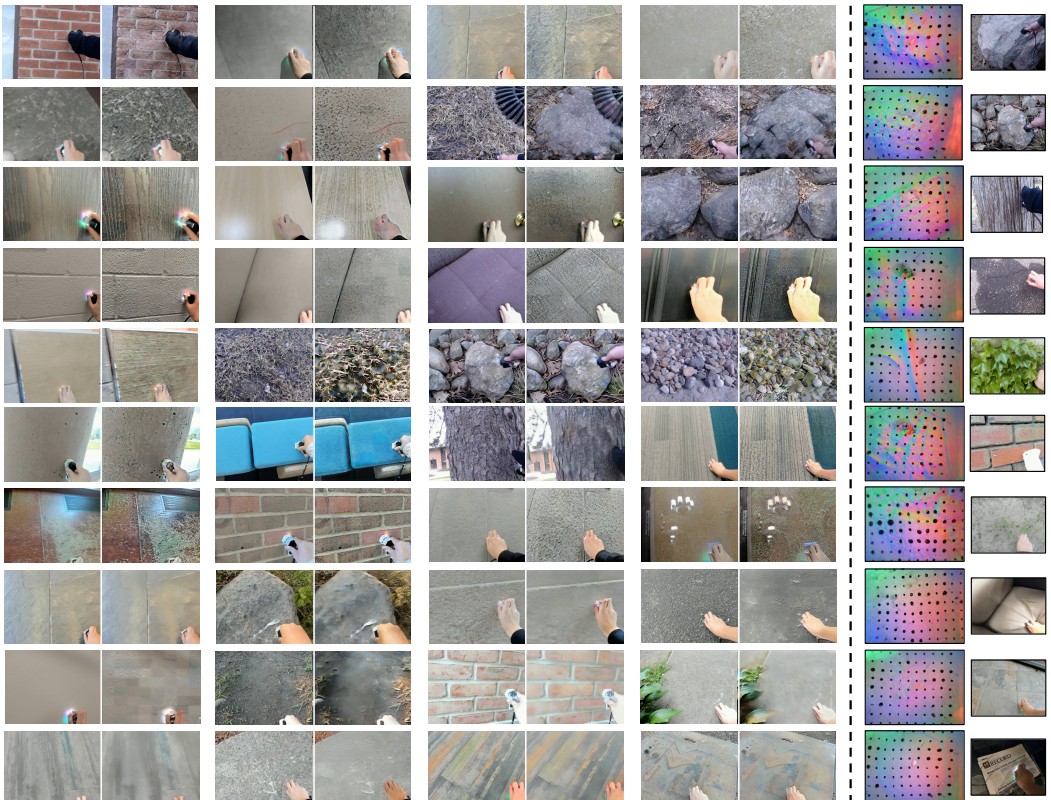

Figure 7: More visualizations of our model on tactile-driven image stylization. For each row, we show an input image (left) and the manipulated image (to its right) obtained by stylizing with a given tactile input (right side). For reference, we also show the image that corresponds to the tactile example at rightmost (not used by the model). *Zoom in for better view.*

## H Details for multimodal future touch prediction

**Overall Architecture** Following [23], our model adopts widely-used residual network from [68] while replacing the 2D convolution to 3D convolution, which utilizes a encoder-decoder architecture. To adapt for multimodal prediction, we introduce two encoders for tactile inputs and visual inputs with identical structures but different weights. Then we concatenate these features along the channel and feed them into the decoder consisting of transposed convolution layers, similar to the architecture of tactile-driven stylization.

**Training details** For the video prediction task, we train our model using Adam Optimizer with the learning rate of $2 \times 10^{-4}$ for all experiments. We utilize the batch size of 8 on 4 Nvidia 2080-Ti GPUs and train for 30 epochs. We initialize the weights from a Gaussian distribution with the mean 0 and std of 0.02. To obtain multi-frame prediction, we recursively feed our output images back to the original model. During this process, the loss are backward through the entire chain of recursive functions and gradients are accumulated, following [23, 44].

**Evaluation Metrics** Following [23], we adopt three evaluation metrics: MAE, SSIM and LPIPS. Structural similarity (SSIM) is a similarity metric to quantify image quality degradation. The higher the SSIM, the better the generated frame. Learned Perceptual Image Patch Similarity (LPIPS) measures the distance between image patches. The lower the LPIPS, the higher the similarity.

# I    Datasheet

## Motivation

**Q1.    For what purpose was the dataset created?**

**Answer:**    The goal of this dataset is to provide training data for multimodal learning systems that learn to associate the sight of objects with their corresponding tactile data (i.e., how they "feel"). In contrast to previous efforts, our dataset contains a large number of in-the-wild recordings from indoor and outdoor scenes.

**Q2.    Who created this dataset (e.g., which team, research group) and on behalf of which entity (e.g., company, institution, organization)?**

**Answer:**    Six researchers at the University of Michigan and Carnegie Mellon University (affiliated as of 2022) have created the dataset: Fengyu Yang, Chenyang Ma, Jiacheng Zhang, Jing Zhu, Wenzhen Yuan and Andrew Owens.

**Q3.    Who funded the creation of the dataset? If there is an associated grant, please provide the name of the grantor and the grant name and number.**

**Answer:**    Our dataset is funded in part by Cisco Systems and The University of Michigan.

**Q4.    Any other comments?**

**Answer:**    No.

## Composition

**Q5.    What do the instances that comprise the dataset represent (e.g., documents, photos, people, countries)?**

**Answer:**    Each instance is a visuo-tactile image pair containing the visual image and its corresponding tactile signal, i.e. the result of someone pressing the object with a GelSight tactile sensor.

**Q6.    How many instances are there in total (of each type, if appropriate)?**

**Answer:**    There are in total approximately 246k visuo-tactile image (frame) pairs of about 13.9k touches in our dataset.

**Q7.    Does the dataset contain all possible instances or is it a sample (not necessarily random) of instances from a larger set?**

**Answer:**    Yes. We have provided the full dataset.

**Q8.    What data does each instance consist of? "Raw" data (e.g., unprocessed text or images) or features?**

**Answer:**    The raw data consists of videos recorded by human collectors and the corresponding tactile videos. The RGB videos and tactile videos are synchronously recorded, and compressed with a video codec.

**Q9.    Is there a label or target associated with each instance?**

**Answer:**    Yes. We label all frames where human are probing an object with its material label.

**Q10.    Is any information missing from individual instances?**

**Answer:**    No.

**Q11.    Are relationships between individual instances made explicit (e.g., users' movie ratings, social network links)?**

**Answer:**    Since we walk through scenes, recording objects around us, the objects in a video are close in space. Tactile signals from the same materials or objects are likely to be similar.

**Q12.    Are there recommended data splits (e.g., training, development/validation, testing)?**

**Answer:** As illustrated in the main text, different tasks require different train/val/test splits. In general, to avoid having the same (or nearly the same) images appear in both training and test set, we recommend splitting the dataset by video (rather than by touch or by frame). We will provide the splits used in our experiments.

**Q13.** **Are there any errors, sources of noise, or redundancies in the dataset?**

**Answer:** It is a challenging task to infer the material according to the RGB images. We have at least 5 people label each image, though still possible to have some images correctly labeled for its material category.

**Q14.** **Is the dataset self-contained, or does it link to or otherwise rely on external resources (e.g., websites, tweets, other datasets)?**

**Answer:** The data is self-contained.

**Q15.** **Does the dataset contain data that might be considered confidential (e.g., data that is protected by legal privilege or by doctor-patient confidentiality, data that includes the content of individuals non-public communications)?**

**Answer:** No.

**Q17.** **Does the dataset relate to people?**

**Answer:** No.

## Collection Process

**Q18.** **How was the data associated with each instance acquired?**

**Answer:** The data is directly collected by two people (authors) walking through a variety of environments, probing objects with tactile sensors and simultaneously recording their actions on videos.

**Q19.** **What mechanisms or procedures were used to collect the data (e.g., hardware apparatus or sensor, manual human curation, software program, software API)?**

**Answer:** We collect our dataset using a RGB camera and a GelSight tactile sensor. Details of the hardware is illustrated in the main text.

**Q20.** **If the dataset is a sample from a larger set, what was the sampling strategy?**

**Answer:** No, the dataset is not a sample from a larger set.

**Q21.** **Who was involved in data collection process (e.g., students, crowd-workers, contractors) and how were they compensated (e.g., how much were crowd-workers paid)?**

**Answer:** Our dataset is collected by authors of this paper.

**Q22.** **Over what timeframe was the data collected? Does this timeframe match the creation timeframe of the data associated with the instances (e.g., recent crawl of old news articles)?**

**Answer:** The dataset is collected across the winter, spring and summer (February 2022 to June 2022). The objects in our dataset are taken from scenes at the specific time (and season) in which the data was collected.

**Q23.** **Were any ethical review processes conducted (e.g., by an institutional review board)?**

**Answer:** Our dataset only contains natural scenes, with no humans subjects (including no humans on screen). It therefore does not qualify as human subjects research.

**Q24.** **Does the dataset relate to people?**

**Answer:** No.

## Preprocessing, Cleaning, and/or Labeling

**Q25.     Was any preprocessing/cleaning/labeling of the data done (e.g., discretization or bucketing, tokenization, part-of-speech tagging, SIFT feature extraction, removal of instances, processing of missing values)?**

**Answer:**     Yes. We collect our raw data in the format of RGB and GelSight videos. To facilitate training and downstream tasks, we preprocess the raw videos by converting them into frames, detecting the frames within the press, and label the pressed frames by their material. Detailed description are in the *Dataset* section of the main text.

**Q26.     Was the "raw" data saved in addition to the preprocessed/cleaned/labeled data (e.g., to support unanticipated future uses)?**

**Answer:**     Yes. We save the original videos for unanticipated future uses of other tasks.

**Q27.     Is the software used to preprocess/clean/label the instances available?**

**Answer:**     Yes. The source code to extract frames is available on our webpage.

**Q28.     Any other comments?**

**Answer:**     No.

## Uses

**Q29.     Has the dataset been used for any tasks already?**

**Answer:**     Yes. As illustrated in the main text, we apply our dataset to a variety of multimodal learning tasks. First, we learn tactile features through self-supervised learning, by training a model to associate images with touch. Secondly, we use our dataset to perform material classification task via GelSight Images. Thirdly, we propose a novel task of *tactile-driven image stylization*: making an image "feel more like" a given tactile input. Finally, we study multimodal models for future touch prediction: predicting future frames of a touch sensor's recording, given both visual and tactile signals.

**Q30.     Is there a repository that links to any or all papers or systems that use the dataset?**

**Answer:**     We do not have a repository to record all papers using our dataset. However, we can track these papers via Google Scholar.

**Q31.     What (other) tasks could the dataset be used for?**

**Answer:**     Our dataset is potentially suitable for tasks that require visual, tactile, or visuo-tactile understanding, such as visual-tactile image translation, shape/hardness estimation, etc.

**Q32.     Is there anything about the composition of the dataset or the way it was collected and preprocessed/cleaned/labeled that might impact future uses?**

**Answer:**     Our dataset was mainly collected in one geographic location (near University of Michigan's campus). Consequently, the data we recorded may not generalize to all spaces. The use of humans in the data collection process also potentially introduces bias, which differs from "robotic" or "virtual data" bias. It was also recorded by a relatively small number of human collectors. The way that they interacted with the objects may therefore not be fully representative.

**Q33.     Are there any tasks for which the dataset should not be used?**

**Answer:**     Our dataset is designed for visuo-tactile learning tasks. It may be not appropriate for tasks outside this domain.

**Q34.     Any other comments?**

**Answer:**     No.

**Q35.     Will the dataset be distributed to third parties outside of the entity (e.g., company, institution, organization) on behalf of which the dataset was created?**

**Answer:**     Yes. Our dataset is publicly available.

**Q36.  How will the dataset will be distributed (e.g., tarball on website, API, GitHub)**

**Answer:**    Our dataset contains a link to a Google Drive directory that contains all of the raw videos, a "`extract_frame.py`" file to convert videos into frames and a separate "`label.txt`" file containing all material labels for frames within the press (See B for more details).

**Q37.  When will the dataset be distributed?**

**Answer:**    We have currently provided all raw data, including videos, tactile recordings, labels, and code. Our dataset will be officially released starting by October 2022 (e.g., in an easy-to-download format and with full documentation).

**Q38.  Will the dataset be distributed under a copyright or other intellectual property (IP) license, and/or under applicable terms of use (ToU)?**

**Answer:**    Our dataset is distributed under the license of CC BY.

**Q39.  Do any export controls or other regulatory restrictions apply to the dataset or to individual instances?**

**Answer:**    No.

**Q40.  Any other comments?**

**Answer:**    No.

## Maintenance

**Q41.  Who will be supporting/hosting/maintaining the dataset?**

**Answer:**    Our dataset is currently hosted on a public Google Drive directory. We will also mirror the dataset using a web server provided by The University of Michigan, so that it will be available indefinitely.

**Q42.  How can the owner/curator/manager of the dataset be contacted (e.g., email address)?**

**Answer:**    The email of authors of our dataset is available on the project webpage.

**Q43.  Is there an erratum?**

**Answer:**    No. If we notice errors in the future, we will put them in an erratum.

**Q44.  Will the dataset be updated (e.g., to correct labeling errors, add new instances, delete instances)?**

**Answer:**    There is no routine update plan for our dataset. To correct labeling errors, please contact authors of our dataset.

**Q45.  If the dataset relates to people, are there applicable limits on the retention of the data associated with the instances (e.g., were individuals in question told that their data would be retained for a fixed period of time and then deleted)?**

**Answer:**    No. Our dataset is not related to people.

**Q46.  Will older versions of the dataset continue to be supported/hosted/maintained?**

**Answer:**    No. We only maintain the latest dataset unless there is a significant update.

**Q47.  If others want to extend/augment/build on/contribute to the dataset, is there a mechanism for them to do so?**

**Answer:**    We have provided information about how the data was collected, including the sensors and the dataset collection procedure. Thus, those who want to collect similar data can easily do so.

