# OpenReview forum: "Touch and Go: Learning from Human-Collected Vision and Touch"
_NeurIPS.cc/2022/Track/Datasets_and_Benchmarks — NeurIPS 2022 Datasets and Benchmarks _

### Official Review · Reviewer_29Lw · 2022-07-10
**A thorough database for touch and sight association tasks, but it is questionable how much benefit the database provides to the tasks.**

**Rating:** 3
**Confidence:** 4
**Clarity:** The paper is well written.

**Strengths:**

The dataset introduces more diversity than prior databases. It also introduces a view of a person probing objects, which keeps more information than databases with only information related to the tactile sensor.

**Weaknesses:**

It is unclear how the dataset benefits from collecting more surfaces and keeping video of humans probing objects. The authors also claim that the dataset can help understand material properties, but this is not well proven in the paper.

**Additional Feedback:**

Adding more objects and multi-modal information to the dataset is great, but it should be carefully considered how these help with learning tasks. Also, the dataset is more likely to be used in robot manipulation tasks, so it may be helpful to find out what information is essential to the tasks and then adjust the dataset.

**Correctness:**

The authors argue that many aspects of the visuo-tactile world are currently best learned by observing physical interactions performed by humans. This is demonstrated by adding more material to the dataset and adding videos of human probing objects, but the experiments in Chapter 5 show only minor improvement by involving the video in the learning tasks.

Another argument is that a dataset with more diversity can help with self-supervised learning and tasks that require an understanding of material property, but experiments in Chapter 5 lack comparison with more or less diversity during learning problems, and the understanding of material property is demonstrated by material classification, which has two major problems: the accuracy is only around 35%, and features of materials(roughness, hardness, etc.) are not extracted.

**Documentation:**

Overall, the documentation is sufficient to use. However, the project pages indicate that each raw video folder in the Dataset folder consists of six items, but only four items are provided in the folder, as video_frame and gelsight_frame are missing. The documentation of how to use extract_frame is also missing. The suffix of time1.npy and time2.npy is mistakenly documented as time1.mp4 and time2.mp4.

**Ethics:**

The dataset mainly consists of objects and should not have ethical issues. The license of the dataset is CC BY 4.0.

**Relation To Prior Work:**

It is clearly discussed how this work differs from the preivous contribution in Table 1.

**Summary And Contributions:**

The paper introduces a database with videos of a webcam and gelsight camera output captured when a person probes objects in natural environments with a tactile sensor. It is more diverse than existing databases, and provides multi-modal information, but how these benefits touch and sight association tasks are unclear.

The main contribution is the dataset with labels of touching / not touching and different materials, which can be used for visuo-tactile learning problems.

---

> ### Author Response · Authors · 2022-08-11
> **Response to Reviewer 29Lw**
>
> Thank you for the thorough comments.
>
> **Q6.1 Predicting other material properties**
>
> A6.1: Following the suggestion, we assigned binary hard/soft and smooth/rough labels to each example. We will describe the annotation process in a revision; it resembles [48]. On the task of predicting these “low level” material labels from touch, we again found that our self-supervised features performed much better than other feature sets:
> |Dataset|Pretraining Method|Hard/soft Acc|Rough/Smooth Acc|
> |-|-|-|-|
> |ImageNet|Supervised|72.3|76.3|
> |VisGel|Visuo-tactile CMC|69.4|70.4|
> |Object Folder 2.0|Visuo-tactile CMC|66.1|65.9|
> |Ours|Visuo-tactile CMC|**77.3**|**79.4**|
> We’ll add these in a revision.
>
> **Q6.2 Low accuracy for material classification**
>
> A6.2: The reason our features get 35.5% accuracy is that this is a very challenging prediction problem. We use a linear classifier to solve a 20-way material classification task, using labels that are annotated from vision, not touch (following [48]). We emphasize that the standard evaluation metric is the *relative* performance of different features, following other self-supervised linear classification evaluations (e.g. [48] and [Zhang et al., “Colorful Image Colorization”, 2016]). On this metric we see large differences between features. E.g. ImageNet features (widely used for GelSight) get only 31.2% accuracy. If absolute performance were a concern, we note that our experiments predicting “low level” material properties (Q6.1) show much higher accuracy.
>
> Interestingly, we found that this task was hard for humans as well. To help put the performance into context, we had 3 authors try to solve it using a 100-sample test set. They had 25.3% accuracy (and online workers were nearly at chance level), perhaps due to the challenge of interpreting tactile images without extensive experience.
>
> **Q6.3 Evidence of understanding material properties**
>
> A6.3: Beyond the material classification task, there are two other lines of evidence supporting the fact that visuo-tactile models learn material properties on our dataset: (1) The new “low level” material property prediction task (Q6.1). (2) The tactile-driven stylization, which shows that the model can change a visually perceived material based on touch (quantitative eval. on Tab. 3). Qualitatively, we see in Fig. 3 that the model transfers the roughness (row 1-3) and smoothness (row 10) of one material to another.
>
> **Q6.4 Benefits of dataset diversity**
>
> A6.4: Following the reviewer’s suggestion, we evaluate the effect of diversity. We train our self-supervised learning model with different numbers of material categories, while keeping the total number of training examples constant (the test set is also constant). Diverse datasets lead to much better downstream performance:
> |Num. Categories|Material Acc|Hard/Soft Acc|Rough/Smooth Acc|Robotic Grasping|
> |-|-|-|-|-|
> |5|23.9|67.3|65.3|62.3|
> |10|29.1|71.2|66.8|66.5|
> |15|32.5|73.3|74.7|70.8|
> |20|**34.2**|**75.6**|**78.6**|**73.8**|
>
> We’ll include this in a revision.
>
> **Q6.5 Minor Improvements in future touch prediction**
>
> A6.5: In fact, the improvement from adding vision to future-touch prediction is significant. For context, it is roughly on par with the gap between different loss functions or architectures on standard video prediction datasets, as shown in Table 2 of [24], even though our only change is to add visual data.
>
> **Q6.6 “the dataset is more likely to be used in robot manipulation tasks, so it may be helpful to find out what information is essential to the tasks and then adjust the dataset.**
>
> A6.6: We are puzzled by this suggestion, since a key motivation of our dataset is to *support work outside of the robotic manipulation field*. Robotics has been the dominant focus of visuo-tactile sensing research today, yet the possible uses for this data is much broader. We see our work as an important step toward supporting these areas of research.
>
> Moreover, it is not clear how to capture “what information is essential to the tasks”. In many areas of machine perception, researchers have had a great deal of success learning general-purpose representations through self-supervision with diverse datasets (e.g., language models and internet text, visual models and ImageNet). In fact, we found our self-supervised features transfer better to a **robotic grasping** task than those of VisGel, a robotic-grasping dataset (supp Tab. 1). This supports the idea that we may be able to learn useful representations from our dataset, without need to tailor collection to one domain.
>
> **Q6.7 Documentations**
>
> A6.7: We thank the reviewer for examining the data! The video_frame and gelsight_frame directories are created *after* running extract_frame.py (supp. L17-20). While extract_frame.py is discussed in the supplement, we would be happy to add more documentation in the code itself. Finally, we thank the reviewer for catching typos in time1.npy and time2.npy. We have updated the webpage to fix these.

---

### Official Review · Reviewer_GEAi · 2022-07-19
**Valuable new dataset with three interesting applications on visuo-tactile learning**

**Rating:** 7
**Confidence:** 4
**Clarity:** Yes, the paper is generally well writ…

**Strengths:**

- This paper proposes an interesting dataset that associates sight with touch. Compared to prior datasets that collect tactile data in controlled lab settings with limited objects or in simulation, this dataset collects real-world tactile data from natural environments.

- Three interesting tasks have been performed to demonstrate the usefulness of the collected dataset: including 1) self-supervised vsual-tactile feature learning, 2) tactile-driven image stylization, and 3) predicting future frames of a tactile signal from visuo-tactile inputs.

- Table 1 nicely summarizes the advantages of the proposed dataset. The new dataset is more human-centered, and it contains more diverse objects in diverse environments.

**Weaknesses:**

- It is motivated that one of the key advantages of this dataset is that it's human-centered, and "the work is a step toward human-collected multimodal data collection...., physically interacting with the world". While the environments where the data are collected from are indeed pretty diverse, the way to collect the data is still not how we humans touch objects in the world (with fingers). The data collection is done with a GelSight sensor. As shown in the qualitative illustrations, a human hand is grasping the sensor to press on different surfaces. It would to add some discussions on this. And why human-collected dataset is better?

- The markers (black dots) seem very noisy and not very well-calibrated. Some are large, and some are small. I wonder if there is any reason for this.

**Additional Feedback:**

- Minor typo, L105, moveed -> moved

**Correctness:**

Yes, the dataset is constructed in a sound way. One question/concern is the quality of the tactile data (see weakness on the question on markers).

**Documentation:**

Yes, the paper and the supplementary materials have nicely documented the data collection process and the details of the dataset.

**Ethics:**

Not to the knowledge of this reviewer.

**Relation To Prior Work:**

Yes, the paper clearly discusses its relation to prior work and compares with prior datasets in Table 1.

**Summary And Contributions:**

The paper introduces a new dataset that contains both the visual and the tactile modalities for visuo-tactile learning. Different from the existing datasets that also contain tactile data, the new dataset is collected in natural environments with diverse objects. Experiments on three tasks have demonstrated the usefulness of the dataset.

---

> ### Author Response · Authors · 2022-08-16
> **Response to Reviewer GEAi**
>
> We thank the reviewer for the thorough comments.
>
> **Q5.1 It is motivated that one of the key advantages of this dataset is that it's human-centered, and "the work is a step toward human-collected multimodal data collection...., physically interacting with the world". While the environments where the data are collectjued from are indeed pretty diverse, the way to collect the data is still not how we humans touch objects in the world (with fingers). The data collection is done with a GelSight sensor. As shown in the qualitative illustrations, a human hand is grasping the sensor to press on different surfaces. It would to add some discussions on this. And why human-collected dataset is better?**
>
> A5.1: The major advantage of human data collectors is that they can acquire much more *diverse* data. Humans can easily access large numbers of objects and scenes. By contrast, having a robot navigate to the objects in our dataset, reach them with its end effector, and safely physically interact with them would be extremely difficult and time consuming for robots; each subtask is an open research area. As a consequence, our dataset contains approximately 3.97x as many object instances as the second-largest dataset.
>
> Touching objects with GelSight is similar to pressing a single finger to the object. This captures the "core problem" of correlating how an object looks with how it feels. Going beyond this and having humans manipulate objects with their hands would indeed be an interesting future direction. However, the problem of obtaining high-quality tactile data during natural hand interactions is a challenging open problem in sensor design, and it is not clear how such data would be recorded. We will clarify this in a revision.
>
> **Q5.2 The black dots on Gelsight seem very noisy and not very well-calibrated. Some are large, and some are small.**
>
> A5.2: These black dots are "markers" that are physically embedded within the GelSight’s elastomer gel. These were proposed in [Yuan et al., "GelSight: High-Resolution Robot Tactile Sensors for Estimating Geometry and Force", 2017] as a way of measuring the full range of forces in a touch, particularly shear force, and are a standard part of GelSight sensors. We refer the reviewer to examples from other GelSight papers [1,2,3] that contain similar variation in marker size, which is typical with this sensor. We will clarify this in a revision to make the paper more accessible.

---

### Official Review · Reviewer_DMKc · 2022-07-21
**A human centric visuo-tactile data collection but lacks standardrization**

**Rating:** 5
**Confidence:** 3
**Correctness:** Please see the above section in Weakn…

**Strengths:**

The visuo-tactile data has the following benefits:
- rigid object data can be collected by human
- probe objects in the wild (both indoors and outdoors)
- a wider range of objects
- a large quantity of data


**Weaknesses:**

- Tactile data collection lacks standardisation
The sequence of exploration procedures (EPs) that Lederman and Klatzky addressed in their experiments can have an effect on the tactile data collected from tactile sensors. The benefits of human-collected tactile data are outlined in the paper whereas the advantage of robotic data collecting remains unknown.  During the data collection, however, the **variable control** is unclear. The pressure or force was not indicated during the data collection process and was only mentioned in the paper as ''*do not perform actions with consistent force*''. This can bring large bias during collection, for example, the leather tactile data shown in the paper is collected by pressing the sofa while synthetic fabric by a cushion, both of them have foam cushion inside, without a constant force, the results can be varied and in reproducible. Lack standardisation could impede the ability of future researchers to continue building the datasets as well. A tactile sensor with a force sensor that indicates a force or force range could be valuable and help with standardisation here.

- The sensor might introduce a data collection bias
All data collected in this paper use the same sensor, it is not clear what the potential bias and limitations are in this sensor and the authors did not present a study on that. In addition, potential users with their own tactile sensors are very likely to suffer from the data bias because of the difference in sensing devices.

- Video recording
The human-centric view has its novelty. However, according to the image in the paper, these recordings may have **tracking registration errors**. In other words, neither the distance between the camera and the object nor the distance between the object and the sensor are constant. A focal point for the sensor in the video recording may provide a solution.

**Additional Feedback:**

I do not have any additional feedback.

**Clarity:**

The paper is well-presented with a clear comparison to some of the existing work. As mentioned previously, it would be nicer if the authors can elaborate on their tactile sensor and the potential limitations that this sensor might have.

**Documentation:**

The dataset is well documented on its github.io and has descriptive guidance for users.

**Relation To Prior Work:**

When compared with other visuo-tactile datasets, the author has simply categorised the collection methods as human or robot, the collection methodology is unclear. According to related work, from my understanding, the task of the human collection can refer to grasping by a robot (not clear), however, the validation part jumps to map with object manipulation. In addition, the robotic manipulation tasks are unclear in the related work. Besides the data collection method comparison, the author could also consider adding tactile sensor comparison as well.


**Summary And Contributions:**

This paper presents a new dataset that helps to understand the material properties of objects in real-life. The authors use human data collectors to probe objects in the wild (both indoors and outdoors) and collect both video and information received on tactile sensors. The proposed dataset is an interesting one and has the potential to be utilized in many downstream tasks, however, its lack of standardization and potential bias in the data collection method might limit its use.

---

> ### Author Response · Authors · 2022-08-11
> **Response to Reviewer DMKc**
>
> Thank you for the thorough comments.
>
> **Q4.1 Standardization**
>
> A4.1: While humans are less consistent data collectors than robots, we think that this concern misses a key motivation for our work: obtaining diverse training data for self-supervision. In other multimodal learning fields, such as vision-and-language, datasets are far less standardized (e.g, internet images/text). Yet they have proven extremely effective for feature learning, arguably because the diversity of the data leads to better transfer ability. Lack of standardization can also be an asset, since it forces models to generalize (e.g., they cannot rely on a certain force or camera pose). In fact, we found that our features transferred better to a robotic grasping task than those of a model trained on *another* robotic grasping dataset (see Q1.2 to L7iv), providing evidence that models trained on our dataset indeed learn useful features.
>
> **Q4.2 Force not indicated during the data collection**
>
> A4.2: We considered measuring force with a separate FSR sensor (and experimented with one in an earlier iteration of the project). However, this significantly complicates data collection, while giving information that the **GelSight already provides implicitly**. We note that there has been extensive work on estimating forces from GelSight, such as [Yuan et al., "GelSight: High-Resolution Robot Tactile Sensors for Estimating Geometry and Force", 2015]. We also emphasize that the paper provides examples of many visuo-tactile learning tasks that can be solved without direct force estimates.
>
> **Q4.3 Possible "bias during collection" without force control**
>
> A4.3: Our experiments sample GelSight frames for a range of times, from the start to end of each press. They thus cover the full range of forces and deformations (barely touching to fully pushing the object). Our benchmarks also extract *static* GelSight frames, reducing the ability to use force as a cue. We’ll clarify this in a revision.
>
> **Q4.4 Leather sofa example and reproducibility**
>
> A4.4: We first refer the reviewer to Q6.1 of R29Lw, where we predict “low level” material properties that may not be as influenced by the amount of deformation (e.g., roughness). The task of predicting the visually-indicated material of a leather sofa from touch (which we assume the reviewer is referring to) would indeed be ambiguous in the initial stages of the press. This would make it a difficult example, but this is not unexpected: cross-modal tasks regularly contain such examples (e.g., in image captioning). Moreover, it is quite *reproducible*: the label is well-defined (obtained from vision), and the GelSight inputs are static frames sampled with different amounts of deformation (see above).
>
> **Q4.5 “Single tactile sensor may introduce bias”**
>
> A4.5: We are surprised by this comment, since single-sensor datasets are extremely common, not only in touch sensing but in other areas of machine perception (e.g., lidar and RGB-D datasets). We use GelSight because it is a very popular and information-rich sensor, used by much of “state of the art” in visuo-tactile sensing (e.g., [8,9,23,40]).  These papers *also use GelSight as their sole tactile sensor*, including the very recent *visuo-tactile dataset paper*, Object Folder 2.0. Finally, using GelSight allows us to directly transfer our self-supervised representations to other works (Sec 5.1).
>
> **Q4.6 “Tracking registration” errors**
>
> A4.6: We admit to being confused by this question. Having the camera/sensor/objects be at fixed distances (our interpretation of the question) would likely hurt the transferability of the self-supervised features. Varying these has the advantage of forcing prediction models to generalize to a variety of different scene configurations, and is a standard collection approach (e.g., [48]). Since our dataset is also an *egocentric video dataset*, where the person moves between objects, it is not clear how the proposed idea could be implemented in practice. We hope that the reviewer can clarify their question if we did not interpret it correctly.
>
> **Q4.7 Definition of “human-collected” and “robot-collected”**
>
> A4.7: “Human” means that objects were probed with a touch sensor by humans; “robot” means that the robot did this instead. This distinction also is synonymous with lab-collected vs. in-the-wild, since the robots are stationary. Perhaps the source of the reviewer’s confusion stems from the fact that self-supervised features we *learn* from human-collected data are subsequently *transferred* to robot-collected datasets. We’ll revise the related work and Table 1 to clarify this.
>
> **Q4.8 Besides the data collection method comparison, the author could also consider adding tactile sensor comparison as well**
>
> A4.8: We assume this refers to Table 1 (and the related work), which we’ll update this in a revision. All datasets use GelSight, except for ProtoPack (which uses several sensors) and Object Folder 1.0 (which uses the similar DIGIT).

---

### Official Review · Reviewer_RMEP · 2022-07-27
**Generally good but having doubts on data collection and visualization**

**Rating:** 7
**Confidence:** 3
**Correctness:** I believe the claims in the submissio…
**Clarity:** The paper is generally well-written.

**Strengths:**

1. Given the drawbacks of robot-centric and simulation-based methodology for visuo-tactile data collection, the authors proposed to collect human-centric data using tactile sensors and ego-centric camera, which is reasonable.
2. The newly proposed dataset is much more diverse than previous efforts.
3. The proposed tactile-driven image stylization is interesting and inspiring.
4. Results on downstream tasks show the promising potential of the dataset.

**Weaknesses:**

I have two minor concerns.
1. The "approximately egocentric" video recording seems a little weird. The adopted camera (according to the photo in the supplementary materials) seems to require acceptable efforts to be made wearable. This makes the current recoding a little clumsy. I'm also having doubts about the stability of the videos recorded. Therefore, I'm curious about the motivation of the "approximately egocentric" video.
2. The visualization of tactile sensors is a little confusing. I'm having trouble understanding the meaning of the visualizations, e.g., the meaning of the dots and the colors.

**Additional Feedback:**

-

**Documentation:**

I believe the documentation is sufficient.

**Ethics:**

I have no such concerns.

**Relation To Prior Work:**

The authors clearly discussed its relation to prior works.

**Summary And Contributions:**

To address the lack of sight-touch correlation data, the authors proposed a multi-modal visuo-tactile dataset.
Instead of being robot-centric like prior efforts, with wearable tactile sensors and egocentric camera, human-centric data involving diverse objects and scenes are collected.
The dataset empowers multiple tasks, including tactile feature prediction, newly proposed tactile-driven image stylization, and future touch prediction.

---

> ### Author Response · Authors · 2022-08-16
> **Response to Reviewer RMEP**
>
> Thank you for the thorough comments.
>
> **Q3.1 The "approximately egocentric" video recording seems a little weird. The adopted camera (according to the photo in the supplementary materials) seems to require acceptable efforts to be made wearable. This makes the current recording a little clumsy. I'm also having doubts about the stability of the videos recorded. Therefore, I'm curious about the motivation of the "approximately egocentric" video.**
>
> A3.1: Thanks for the helpful suggestion. It would indeed be relatively straightforward to make the entire recording setup wearable, and we think that this would be a great direction in future work. We opted to have two data collectors for convenience, because the combination of different devices (computer, camera, GelSight) can be hard for a single person to operate simultaneously, and because the computer is somewhat bulky. In particular, we found it helpful to have one operator carry the computer, providing the other operator with a real-time display of the images and tactile data. In this way, they can ensure that the objects they are probing appear approximately in the center of the recorded images and increase the stability of the recording. We will clarify this in a revision.
>
> **Q3.2 The visualization of tactile sensors is a little confusing. Hard to understand the meaning of the dots and the colors.**
>
> A3.2: To visualize the tactile data, we show the raw video recording from the GelSight sensor, following standard practice e.g. [Yuan et al., "GelSight: High-Resolution Robot Tactile Sensors for Estimating Geometry and Force", 2017]. As noted in the paper, we also apply contrast enhancement to better convey small shape variations. The GelSight uses a camera to record an elastomer gel, which is illuminated by colored LED lights. When the sensor is pressed against an object, the gel deforms, which results in changes to the reflected illumination. The color conveys the surface normal of the object being touched, similar to photometric stereo. These black dots are "markers" that are physically embedded within the GelSight’s elastomer gel. These were proposed in [Yuan et al., "GelSight: High-Resolution Robot Tactile Sensors for Estimating Geometry and Force", 2017] as a way of measuring the full range of forces in a touch, particularly shear force, and are a standard part of GelSight sensors. We will clarify this in the paper.

---

### Official Review · Reviewer_QTFy · 2022-07-28
**Review for Touch and Go**

**Rating:** 5
**Confidence:** 5
**Correctness:** The claims appear to be correct.
**Clarity:** The paper is well written.

**Strengths:**

1. The scale of the proposed dataset is larger than previous ones and the diversity of objects and senarios is enhanced.
2. Various applications are presented on the proposed dataset, including a novel tactile-driven image stylization task.

**Weaknesses:**

1. The functionality of the dataset remains undetermined since the results are not compared with those of other visual-tactile datasets (e.g. for Robotic grasping only ImageNet but no other visual-tactile datasets are compared).
2. Although interesting, the motivation of the proposed tactile-driven image stylization is not clearly stated, and the qualitative results seem trivial.
3. Benchmarking on the proposed dataset is inadequate. For touch prediction only one model is evaluated.

**Additional Feedback:**

No additional feedback.

**Documentation:**

The details of data collection and organization are clear.

**Ethics:**

No ethical concerns.

**Relation To Prior Work:**

Prior works are compared with the proposed dataset and the difference in scale are highlighted.

**Summary And Contributions:**

This paper provide a large visual-tactile dataset with GelSight tactile sensor and camera, containing diverse senarios and objects.Experimental analysis is carried out on self-supervised feature learning, tactile-driven image stylization and future tactile image prediction.

---

> ### Author Response · Authors · 2022-08-21
> **Response to Reviewer QTFy (Part1)**
>
> We thank the reviewer for the thorough feedback.
>
> **Q2.1 No comparison to other visual-tactile datasets**
>
> A2.1: In fact, there is already a direct comparison to other visuo-tactile datasets for the self-supervised learning task and its downstream evaluation  (**supplementary material Table 1**). In these experiments, we learned self-supervised features on the VisGel and Object Folder datasets using CMC, and found that our features significantly outperformed them on downstream tasks. Interestingly, our features obtained the best performance on the robotic grasping dataset of Calandra et al., even though one of the compared datasets (VisGel) is a large, *robot-collected* dataset. This provides further support for the idea that general-purpose representations can be learned from human-collected data. We will move this comparison to the main text, since we feel that it may have been overlooked.
>
> Beyond this experiment, we feel that our comparison to other datasets is quite extensive, especially given that our dataset is very different from prior work. We have: 1) compared the contents of our dataset to other visuo-tactile datasets (main paper Table 1), 2) provided qualitative samples from several datasets (supplement Figure 1), and 3) demonstrated cross-dataset generalization for robotic grasping (main paper Table 2). This arguably goes significantly beyond the cross-dataset comparisons that have been provided by recent visuo-tactile dataset papers, such as Object Folder.
>
> **Q2.2 Motivation for tactile-driven stylization.**
>
> A2.2: Manipulating images to match "tactile prompts" provides another way of evaluating whether models can learn cross-modal visual-tactile associations -- i.e., how an object feels from how it looks and vice versa. In other areas of multimodal perception (perhaps most strikingly in the language-and-vision field) cross-modal image synthesis has provided an important confirmation that models can indeed learn these associations, supplementing "quantitative only" evaluations like feature learning. In our paper, we showed that we can easily adapt a recent audio-visual stylization method [Li et al., "Learning Visual Styles from Audio-Visual Associations", ECCV 2022] to the visuo-tactile domain, by replacing audio with touch, *suggesting that our dataset can indeed support research in cross-modal image synthesis*.
>
> Beyond providing this confirmation that our dataset can be used for cross-modal synthesis (which was our main motivation), tactile-driven image manipulation is an important end in itself, for the reasons outlined in audio-visual stylization work [Li et al., ECCV 2022]. Touch provides complementary information that may not be easily conveyed through other modalities that are commonly used to drive image stylization, such as language and sound. For example, touch can precisely define how smooth/rough a surface ought to be, and express the subtle shape of its microgeometry. We also note a connection to work in computer graphics, where a long-standing goal has been to synthesize images that have specific material properties [Guerrero et al. MatFormer: A Generative Model for Procedural Materials, 2022]. Interestingly, these material properties are captured implicitly via a tactile signal. We will clarify this motivation in a revision, by expanding the introduction and Section 4.2.
>
>
> **Q2.3 “Qualitative results of tactile-driven stylization are trivial”**
>
> A2.3: We admit to being confused by the reviewer's question, since it is somewhat ambiguous; it is not clear what they mean by "trivial". (1) If their concern is about the result’s importance, we refer them to the answer to the previous question. (2) If their concern is about the quality of the results, we note that we have also supplied *quantitative* evaluations (Table 3), which confirm that our model outperforms a variety of baselines. (3) Finally, if the concern is that the magnitude of the image manipulations are too small, we point out that they are similar in magnitude to those in the closest prior work [Li et al., ECCV 2022] (e.g., adding rain or snow to a scene using sound). We hope the reviewer can clarify their question, if we have not fully addressed it.

---

> > ### Author Response · Authors · 2022-08-21
> > **Response to Reviewer QTFy (Part2)**
> >
> > **Q2.4 Other future touch prediction evaluations.**
> >
> > A2.4: Following the reviewer's advice, we have added an additional video prediction method, the popular SVG [Denton et al.  “Stochastic Video Generation with a Learned Prior”, ICML 2018]. We again find that visual information provides an improvement in performance. We have provided an updated table below. We will add this to a revision.
> >
> > Table 1. Time horizon of skipping 3 frames
> > | Method | Modality | L1 | SSIM | LPIPS |
> > |---|---|---|---|---|
> > | SVG | Vision | 0.782 | 0.572 | 0.391 |
> > | Geng et al.[24] | Vision | 0.628 | 0.708 | 0.103 |
> > | SVG | Vision + Touch | 0.757 | 0.602 | 0.368 |
> > | Geng et al.[24] | Vision + Touch | **0.617** | **0.719** | **0.091** |
> >
> > Table2. Time horizon of skipping 5 frames
> > | Method | Modality | L1 | SSIM | LPIPS |
> > |---|---|---|---|---|
> > | SVG | Vision | 0.807 | 0.513 | 0.412 |
> > | Geng et al.[24] | Vision | 0.691 | 0.698 | 0.279 |
> > | SVG | Vision + Touch | 0.762 | 0.546 | 0.397 |
> > | Geng et al.[24] | Vision + Touch | **0.663** | **0.713** | **0.265** |
> >
> > **Q2.5 Amount of benchmarking on the proposed dataset**
> >
> > A2.5: We hope that our extended future touch prediction evaluation (Q2.4) addresses the reviewer’s concern. We note, however, that the goal of this paper is to propose a dataset for a new type of data, “in the wild” vision-and-touch, and to demonstrate its usefulness for a variety of visuo-tactile understanding tasks. It is not a paper that addresses an existing, well-established task, with large numbers of existing methods to compare with. In fact, the future touch prediction task does not have existing methods to directly compare to, so each model in our evaluation required modifications. Given that, we think that the amount of benchmarking is quite extensive. In particular, we compared self-supervised representations from several different datasets and applied them to a downstream task on a robotics dataset (Q1.2 for L7iv), arguably going significantly beyond other recent vision-and-touch dataset papers.

---

### Official Review · Reviewer_L7iv · 2022-07-28
**An large dataset for multimodal visuo-tactile learning.**

**Rating:** 6
**Confidence:** 3
**Clarity:** As far as I am concerned,this paper c…

**Strengths:**

---1) The data are collected by human in an “approximately egocentric” manner,  which is thought to have higher diversity than those data collected by robots.
---2) Compared with previous efforts, this dataset has the largest scale in terms of the number of categories of objects and scences. It also contains the largest number of touches.
---3) Extensive experiments have been done, and the results show that this dataset is effective on different downstream tasks.

**Weaknesses:**

1) the timings of touch onsets are labeled by machine instead of human, which may cause error of labeling.
2) there is no comparison with other visuo-tactile datasets, so it's no clear what benefits this dataset can bring to the downstream tasks.

**Additional Feedback:**

None

**Correctness:**

The dataset is constructed in a sound way. And expriements cover a wide range of tasks. the results show the effectiveness of this dataset. but the comparison with other visuo-tactile datasets are missed.

**Documentation:**

the URL for dataset, and a hosting, licensing and maintenance plan are given.  The codes will be released later.

**Ethics:**

no ethics concerns

**Relation To Prior Work:**

yes. The objects and scenes in this dataset are significantly more diverse than prior efforts, making the data well-suited to tasks that involve
understanding material properties and physical interactions in the wild.

**Summary And Contributions:**

In this work, a new dataset for multimodal visuo-tactile learning called Touch and Go is proposed,  in which human data collectors probe objects in natural environments with tactile sensors, while recording egocentric video.
The objects and scenes in this dataset are significantly more diverse than prior efforts. Experiments on three kinds of tasks have been conducted to demonstrate this dataset’s effectiveness :-1) self-supervised visuo-tactile feature learning, -2) tactile-driven image stylization,
-3) predicting future frames of a tactile signal from visuo-tactile inputs.

---

> ### Author Response · Authors · 2022-08-02
> **Response to Reviewer L7iv**
>
> We thank the Reviewer L7iv for the helpful comments. We have addressed their concerns below:
>
> **Q1.1 The timings of touch onsets are labeled by machine instead of human, which may cause error of labeling.**
>
> A1.1: In fact, our onset annotation process is only semi-automated: the classifier’s predictions are manually corrected by humans (see line 134). In practice, this does not happen often, because the classifier is very accurate. On our test set (2k hand-labeled frames), it obtains 97% accuracy. We will clarify this in a revision.
>
> **Q1.2 There is no comparison with other visuo-tactile datasets, so it's no clear what benefits this dataset can bring to the downstream tasks.**
>
> A1.2: In fact, there is already a direct comparison to other visuo-tactile datasets on downstream tasks (**supplementary material Table 1**). In these experiments, we learned self-supervised features on the VisGel and Object Folder datasets using CMC, and found that our features significantly outperformed them on downstream tasks. Interestingly, our features obtained the best performance on the robotic grasping dataset of Calandra et al., even though one of the compared datasets (VisGel) is a large, *robot-collected* dataset. This provides further support for the idea that general-purpose representations can be learned from human-collected data. We will move this comparison to the main text, since we feel that it may have been overlooked.
>
> Beyond this experiment, we feel that our comparison to other datasets is quite extensive, especially given that our dataset is very different from prior work. We have: 1) compared the contents of our dataset to other visuo-tactile datasets (main paper Table 1), 2) provided qualitative samples from several datasets (supplement Figure 1), and 3) demonstrated cross-dataset generalization for robotic grasping (main paper Table 2). This arguably goes significantly beyond the cross-dataset comparisons that have been provided by recent visuo-tactile dataset papers, such as Object Folder. We would be happy to provide additional comparisons, but it is not clear what these should be. We hope that the reviewer can clarify which specific comparisons they would like to see.

---

> > ### Comment · Reviewer_L7iv · 2022-08-29
> > **Thanks for your response**
> >
> > I will raise the score from 5 to 6.

---

### Author Response · Authors · 2022-08-25
**Revision of the paper**

Dear Reviewers:

We have now posted a revision to our paper, which complements the responses we have made to each of the 6 reviewers over the past 3+ weeks. We highlight the change in red within the revision. We’d like to further emphasize a few key points from our responses:
1. One of the major concerns of both RL7iv and RQTFy was the comparison to other datasets. In fact, such a comparison **already exists** in the supplementary material, which may have been overlooked by reviewers (see [Q1.2 and Q2.1]). We have moved this comparison to the main text. We hope that these two reviewers can comment on whether these existing experiments address their concerns before the end of the discussion period.
2. We showed that visual information improves future touch prediction for an additional model, based on SVG [Denton and Fergus, 2018] (see Q2.4)
3. We added an additional evaluation that tests the learned features’ ability to convey “lower level” material properties: hard vs. soft and smooth vs. rough (see Q6.1)
4. We have evaluated the effect of dataset diversity (see Q6.4).
5. We have reported the accuracy of the classifier that tests whether the GelSight has been pressed (see Q1.1)
6. We discussed concerns about the lack of explicit force recordings and potential bias (see Q4.1).
7. We added additional comparisons of the tactile sensors used by other datasets (see Q4.8).
8. We refined the motivation and related work for tactile-driven stylization (see Q2.2).


We hope that the reviewers will respond to these comments.

---

### Meta-Review · Area_Chair_3VWW · 2022-09-09

**Recommendation:** Accept
**Confidence:** 4

**Metareview:**

Overall, all reviewers see the novelty in the dataset that provides an original new set of tactile data, accounting for a wide range of objects and scene, collected in the wild. The paper is overall well written, clear and easy to follow, and provides a good demonstration of the importance and usefulness of the dataset, going beyond datasets collected by robots in the context of material properties. It also includes initial steps towards benchmarking, although it is not the core contribution of the paper, which lies in the creation of the dataset itself.

Two reviewers recommend acceptance, two stand marginally below acceptance, and one does recommend rejection. The authors have provided detailed responses to all reviewers’ comments and concerns and updated their manuscript. The main concerns of those reviewers sitting on the fence seem to be answered and limitations acknowledged however the element of bias in the data collection was still considered as a possible concern. Based on my reading and accounting for the authors changes, there approach itself is novel and original and hence provides already a relevant contribution to NeuroIPS; while the authors acknowledge limitations and future work avenues. The one reviewer recommending rejection didn’t engage in the discussion but from my reading of both the paper, the comments, and the authors responses, I believe the main points were addressed by the authors and misunderstandings clarified. Overall, I believe the papers offers a novel, non-existing dataset that can inspire interesting future works; all the dataset is shared and presented clearly for others to use.

---

### Decision · Program_Chairs · 2022-09-16

Accept